# CORPUS BASED AMHARIC SENTIMENT LEXICON GENERATION

## ABSTRACT

Sentiment classification is an active research area with several applications including analysis of political opinions, classifying comments, movie reviews, news reviews and product reviews. To employ rule based sentiment classification, we require sentiment lexicons. However, manual construction of sentiment lexicon is time consuming and costly for resource-limited languages. To bypass manual development time and costs, we tried to build Amharic Sentiment Lexicons relying on corpus based approach. The intention of this approach is to handle sentiment terms specific to Amharic language from Amharic Corpus. Small set of seed terms are manually prepared from three parts of speech such as noun, adjective and verb. We developed algorithms for constructing Amharic sentiment lexicons automatically from Amharic news corpus. Corpus based approach is proposed relying on the word co-occurrence distributional embedding including frequency based embedding (i.e. Positive Point-wise Mutual Information PPMI). Using PPMI with threshold value of 100 and 200, we got corpus based Amharic Sentiment lexicons of size 1811 and 3794 respectively by expanding 519 seeds. Finally, the lexicon generated in corpus based approach is evaluated.

**keywords: Amharic Sentiment lexicon , Amharic Sentiment Classification , Seed words**

## 1 INTRODUCTION

Most of sentiment mining research papers are associated to English languages. Linguistic computational resources in languages other than English are limited. Amharic is one of resource limited languages. Due to the advancement of World Wide Web, Amharic opinionated texts is increasing in size. To manage prediction of sentiment orientation towards a particular object or service is crucial for business intelligence, government intelligence, market intelligence, or support decision making. For carrying out Amharic sentiment classification, the availability of sentiment lexicons is crucial. To-date, there are two generated Amharic sentiment lexicons. These are manually generated lexicon(1000) (Gebremeskel, 2010) and dictionary based Amharic SWN and SOCAL lexicons (Neshir Alemneh et al., 2019). However, dictionary based generated lexicons has short-comings in that it has difficulty in capturing cultural connotation and language specific features of the language. For example, Amharic words which are spoken culturally and used to express opinions will not be obtained from dictionary based sentiment lexicons. The word ጉርሻ/"feed in other people with hands which expresses love and live in harmony with others"/ in the Amharic text: "እንደ ጉርሻ ግን የሚያግባባን የለም. . . ጉርሻ እኮ አንዱ ለሌላው የማጉረስ ተግባር ብቻ አይደለም፤ በተጠቀለለው እንጀራ ውስጥ ፍቅር አለ፤ መተሳሰብ አለ፤ አክብሮት አለ።" has positive connotation or positive sentiment. But the dictionary meaning of the word ጉርሻ is "bonus". This is far away from the cultural connotation that it is intended to represent and express. We assumed that such kind of culture (or language specific) words are found in a collection of Amharic texts. However, dictionary based lexicons has short comings to capture sentiment terms which has strong ties to language and culture specific connotations of Amharic. Thus, this work builds corpus based algorithm to handle language and culture specific words in the lexicons. However, it could probably be impossible to handle all the words in the language as the corpus is a limited resource in almost all less resourced languages like Amharic. But still it is possible to build sentiment lexicons in particular domain where large amount of Amharic corpus is available. Due to this reason, the lexicon built using this approach is usually used for lexicon based

sentiment analysis in the same domain from which it is built.

The research questions to be addressed utilizing this approach are: (1) How can we build an approach to generate Amharic Sentiment Lexicon from corpus? (2)How do we evaluate the validity and quality of the generated lexicon? In this work, we set this approach to build Amharic polarity lexicons in automatic way relying on Amharic corpora which is mentioned shortly. The corpora are collected from different local news media organizations and also from facebook news' comments and you tube video comments to extend and enhance corpus size to capture sentiment terms into the generated PPMI based lexicon.

## 2 Related Works

In this part, we will present the key papers addressing corpus- based Sentiment Lexicon generation. In (Velikovich et al., 2010), large polarity lexicon is developed semi-automatically from the web by applying graph propagation method. A set of positive and negative sentences are prepared from the web for providing clue to expansion of lexicon. The method assigns a higher positive value if a given seed phrase contains multiple positive seed words, otherwise it is assigned negative value. The polarity p of seed phrase i is given by: $p_i = p_i^+ - \beta p_i^-$, where $\beta$ is the factor that is responsible for preserving the overall semantic orientations between positive and negative flow over the graph. Both quantitatively and qualitatively, the performance of the web generated lexicon is outperforming the other lexicons generated from other manually annotate lexical resources like WordNet. The authors in (Hamilton et al., 2016) developed two domain specific sentiment lexicons (historical and online community specific) from historical corpus of 150 years and online community data using word embedding with label propagation algorithm to expand small list of seed terms. It achieves competitive performance with approaches relying on hand curated lexicons. This revealed that there is sentiment change of words either positively to negatively or vice-versa through time. Lexical graph is constructed using PPMI matrix computed from word embedding. To fill the edges of two nodes $(w_i, w_j)$, cosine similarity is computed. To propagate sentiment from seeds in lexical graph, random walk algorithm is adapted. That says, the polarity score of a seed set is proportional to probability of random walk from the seed set hitting that word. The generated lexicon from domain specific embedding outperforms very well when compared with the baseline and other variants.

Our work is closely associated to the work of Passaro et al. (2015). Passaro et al. (2015) generated emotion based lexicon by bootstrapping corpus using word distributional semantics (i.e. using PPMI). Our approach is different from their work in that we generated sentiment lexicon rather than emotion lexicon. The other thing is that the approach of propagating sentiment to expand the seeds is also different. We used cosine similarity of the mean vector of seed words to the corresponding word vectors in the vocabulary of the PPMI matrix. Besides, the threshold selection, the seed words part of speech are different from language to language. For example, Amharic has few adverb classes unlike Italian. Thus, our seed words do not contain adverbs.

## 3 Proposed Corpus based Approaches

There are variety of corpus based strategies that include count based(e.g.PPMI) and predictive based(e.g. word embedding) approaches. In this part, we present the proposed count based approach to generate Amharic Sentiment lexicon from a corpus. In Figure 1, we present the proposed framework of corpus based approach to generate Amharic Sentiment lexicon. The framework has four components: (Amharic News) Corpus Collections, Preprocessing Module, PPMI Matix of Word-Context, Algorithm to generate (Amharic) Sentiment Lexicon resulting in the Generated (Amharic) Sentiment Lexicon.

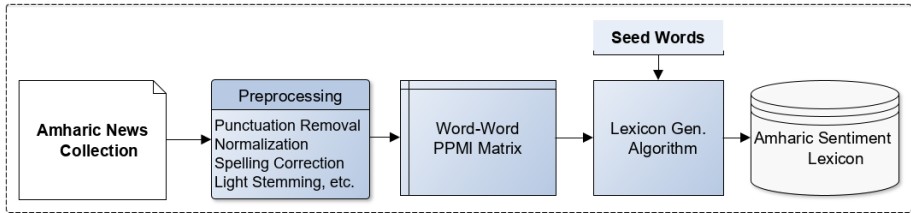

Figure 1: Corpus based Amharic Sentiment Lexicon Building Framework

The algorithm and the seeds in figure 1 are briefly described as follows. To generate Amharic Sentiment lexicon, we follow four major steps:

1. Prepare a set of seed lists which are strongly negatively and positively polarized Adjectives, Nouns and Verbs (Note: Amharic language contains few adverbs (Passaro et al., 2015), adverbs are not taken as seed word). We will select at least seven most polarized seed words for each of aforementioned part-of-speech classes (Yimam, 2000). Selection of seed words is the most critical that affects the performance of bootstrapping algorithm (Waegel, 2003). Most authors choose the most frequently occurring words in the corpus as seed list. This is assumed to ensure the greatest amount of contextual information to learn from, however, we are not sure about the quality of the contexts. We adapt and follow seed selection guidelines of Turney & Littman (2003). After we tried seed selection based on Turney & Littman (2003), we update the original seed words. Sample summary of seeds are presented in Table 1.

2. Build semantic space word-context matrix (Potts, 2013; Turney & Pantel, 2010) using the number of occurrences(frequency) of target word with its context words of window size±2. Word-Context matrix is selected as it is dense and good for building word rich representations (i.e. similarity of words) unlike word-document design which is sparse and computationally expensive (Potts, 2013; Turney & Pantel, 2010).

   Initially, let F be word-context raw frequency matrix with $n_r$ rows and $n_c$ columns formed from Amharic text corpora. Next, we apply weighting functions to select word semantic similarity descriminant features. There are variety of weighting functions to get meaningful semantic similarity between a word and its context. The most popular one is Point-wise Mutual Information (PMI) (Turney & Littman, 2003). In our case we use positive PMI by assigning 0 if it is less than 0 (Bullinaria & Levy, 2007). Then, let X be new PMI based matrix that will be obtained by applying Positive PMI(PPMI) to matrix F. Matrix X will have the same number of rows and columns matrix F. The value of an element $f_{ij}$ is the number of times that word $w_i$ occurs in the context $c_j$ in matrix F. Then, the corresponding element $x_{ij}$ in the new matrix X would be defined as follows:

$$x_{ij} = \begin{cases} PMI(w_i, c_j) \, if \, PMI(w_i, c_j) > 0 \\ 0 \, if \, PMI(w_i, c_j) \ \leq 0 \end{cases} \quad (1)$$

Where, $PMI(w_i, c_j)$ is the Point-wise Mutual Information that measures the estimated co-occurrence of word $w_i$ and its context $c_j$ and is given as:

$$PMI(w_i, c_j) = \log \frac{P(w_i, c_j)}{P(w_i)P(c_j)} \quad (2)$$

Where, $P(w_i, c_j)$ is the estimated probability that the word $w_i$ occurs in the context of $c_j$ , $P(w_i)$ is the estimated probability of $w_i$ and $P(c_j)$ is the estimated probability of $c_i$ are defined in terms of frequency $f_{ij}$.

3. Compute the cosine distance between target term and centroid of seed lists (e.g. centroid for positive adjective seeds, $\overrightarrow{\mu^+_{adj}}$ ). To find the cosine distance of a new word from seed list, first we compute the centroids of seed lists of respective POS classes; for example, centroids for positive seeds S+ and negative seeds S-, for adjective class is given by:

$$\overrightarrow{\mu^+_{adj}}(S^+) = \frac{\Sigma_{w \in S^+}\vec{w}}{|S^+|} \quad \& \quad \overrightarrow{\mu^-_{adj}}(S^-) = \frac{\Sigma_{w \in S^-}\vec{w}}{|S^-|} \tag{3}$$

Similarly, centroids of the other seed classes will be found. Then, the cosine distances of target word from positive and negative adjective seeds of centroids, $\overrightarrow{\mu^+_{adj}}$ and $\overrightarrow{\mu^-_{adj}}$ is given by:

$$cosine(\vec{w_i}, \overrightarrow{\mu^+_{adj}}) = \frac{\vec{w_i} \cdot \overrightarrow{\mu^+_{adj}}}{||\vec{w_i}|| ||\overrightarrow{\mu^+_{adj}}||} \quad \& \quad cosine(\vec{w_i}, \overrightarrow{\mu^-_{adj}}) = \frac{\vec{w_i} \cdot \overrightarrow{\mu^-_{adj}}}{||\vec{w_i}|| ||\overrightarrow{\mu^-_{adj}}||} \tag{4}$$

As word-context matrix x is vector space model, the cosine of the angle between two words vectors is the same as the inner product of the normalized unit word vectors. After we have cosine distances between word $w_i$ and seed with centroid $\vec{w_i}, \mu^+_{adj}$ , the similarity measure can be found using either:

$$Sim(\vec{w_i}, \overrightarrow{\mu^+_{adj}}) = \frac{1}{cosine(\vec{w_i}, \overrightarrow{\mu^+_{adj}})} \quad or \quad Sim(\vec{w_i}, \overrightarrow{\mu^+_{adj}}) = 1 - cosine(\vec{w_i}, \overrightarrow{\mu^+_{adj}}) \tag{5}$$

Similarly, the similarity score, $Sim(\vec{w_i}, \overrightarrow{\mu^-_{adj}})$ can also be computed. This similarity score for each target word is mapped and scaled to appropriate real number. A target word whose sentiment score is below or above a particular threshold can be added to that sentiment dictionary in ranked order based on PMI based cosine distances. We choose positive PMI with cosine measure as it is performed consistently better than the other combination features with similarity metrics: Hellinger, Kullback-Leibler , City Block, Bhattacharya and Euclidean (Bullinaria & Levy, 2007).

4. Repeat from step 3 for the next target term in the matrix to expand lexicon dictionary. Stop after a number of iterations defined by a threshold acquired experimental testing.
The detail algorithm for generating Amharic sentiment lexicon from PPMI is presented in algorithm 1

Algorithm description: The algorithm 1 reads the seed words and generates the merge of expanded seed words using PPMI. Line 1 loads the seed words and assigns to their corresponding category of seed words. Similarly, from line 2 to 6 loads the necessary lexical resources such as PPMI matrix, vocabulary list, Amharic-English, Amharic-Amharic, Amharic-Sentiment SWN and in line 7, the output Amharic Sentiment Lex. by PPMI is initialized to Null. From line 8 to 22, it iterates for each seed words polarity and categories. That is, line 9 to 11 checks that each seed term is found in the corpus vocabulary. Line 12 initializes the threshold by a selected trial number(in our case 100,200,1000, etc.). From line 13 to 22, iterates from i=0 to threshold in order to perform a set of operations. That is, line 16 computes the mean of the seed lexicon based on equation 3 specified in the previous section. Line 17 computes the similarity between the mean vector and the PPMI word-word co occurrence matrix and returns the top i most closest terms to the mean vector based on equation 5. Lines 18-19, it removes top closest items which has different part-of-speech to the seed words. Line 20-21 check the top i closest terms are has different polarity to the seed lexicon. Line 22 updates the PPMI lexicon by inserting the newly obtained sentiment terms. Line 23 returns the generated Amharic Sentiment lexicon by PPMI. Using corpus based approach, Amharic sentiment lexicon are built where it allows

**Input:** PPMI : Word-Context PPMI matrix
**Output:** AM_Lexicon_by_PPMI: Generated Amharic Sentiment Lexicon

1   $seed\_noun^+, seed\_noun^-, seed\_adj^+, seed\_adj^-, seed\_verb^+, seed\_verb^-$   $\leftarrow$ $LoadCorespondingSeedCatagory file$   $PPMI$   $\leftarrow$   $LoadPPMIMatrix file$   $Vocab$   $\leftarrow$ $LoadVocabulary file$   $AmharicEnglishDic$   $\leftarrow$   $LoadAmharicEnglishDictionary file$ $AmharicAmharicDic$   $\leftarrow$   $LoadAmharicAmharicDictionary file$   $AmharicSWN$   $\leftarrow$ $LoadAmharicSentimentSWN file$   $Amharic\_Sentiment\_Lex\_by\_PPMI \leftarrow Null$ **foreach** $seed\_lexicon \in seed\_noun^+, seed\_noun^-, seed\_adj^+, seed\_adj^-, seed\_verb^+, seed\_verb^-$: **do**

2    **foreach** $seed \in seed\_lexicon$: **do**

3      **if** $seed \in Vocab$ **then**

4        Remove Seed from Seed_Lexicon

5    Threshold ←Number of iterations    **foreach** $i \leftarrow 0\ to\ Threshold$: **do**

6      mean_vector ← compute_mean(seed_lexicon) by equation 3
      top_ten_closest_terms ← compute_similarity(mean_vector,PPMI) by equation 4
      **if** $term \in top\_ten\_closest\_terms\ and\ term \in seed\_lexicon$: **then**

7        Remove the term from top_ten_closest_terms list as it is duplicate

8      **if** *Any term in top_ten_closest_terms has different part_speech from seed_lexicon* **then**

9        Remove the term from top_ten_closest_terms list

10      **if** *Any term in top_ten_closest_terms has different polarity from Amharic SWN lexicon* **then**

11        Remove the term from top_ten_closest_terms list

12      Update seed_lexicon by inserting top_ten_closest_terms list

13    AM_Lexicon_by_PPMI ←AM_Lexicon_by_PPMI + seed_lexicon;

algorithm 1: Amharic Sentiment Lexicon Generation Algorithm Using PPMI

finding domain dependent opinions which might not be possible by sentiment lexicon generated using dictionary based approach. The quality of this lexicon will be evaluated using similar techniques used in dictionary based approaches (Neshir Alemneh et al., 2019). However, this approach may not probably produce sentiment lexicon with large coverage as the corpus size may be insufficient to include all polarity words in Amharic language. To reduce the level of this issue, we combine the lexicons generated in both dictionary based and corpus based approaches for Amharic Sentiment classification.

## 4   RESULTS AND DISCUSSIONS

Using corpus based approach, Amharic sentiment lexicon is built where it allows finding domain dependent opinions which might not be possible by sentiment lexicon generated using dictionary based approach. In this section, we have attempted to develop new approaches to bootstrapping relying on word-context semantic space representation of large Amharic corpora.

### 4.1   SEED WORDS

We have manually prepared Amharic opinion words with highest sentimental strength either positively or negatively from three parts- of -speech categories: Adjectives, Nouns and Verbs. We expanded each seed category from which the snapshot seed words are presented in Table 1

Table 1: Selected Seeds for Corpus based Amharic Sentiment Lexicon Expansion

| Part of speech | Negative | Positive |
|---|---|---|
| Adjectives | 99 | 79 |
| Nouns | 145 | 120 |
| Verbs | 36 | 40 |

### 4.2   STEMMING

Amharic is highly morphological language both inflectional and derivational morphology is complex. Thus, without applying stemming, it is not easy to do computational

linguistic tasks like sentiment classification. Amharic roots are very few thousands from which other words are derived. The level of stemming determines whether stemmed terms preserve the semantics of original word or not. So, in this case, we developed light stemmer to reduce surface words into corresponding stems with minimal lose of semantic information. A sample of stemmed words with thier coresponding surface words are presented in Table 2.

## 5 EVALUATION

### 5.1 CORPUS AND LEXICAL RESOURCES

The corpus and data sets used in this research are presented as follows:
i. Amharic Corpus: The size of this corpus is 20 milion tokens (teams from Addis Ababa University et al.). This corpus is used to build PPMI matrix and also to evaluate the coverage of PPMI based lexicon.
ii. Facebook Users' Comment This data set is used to build PPMI matrix and also to evaluate subjectivity detection, lexicon coverage and lexicon based sentiment classification of the generated Amharic Sentiment lexicon. The data set is manually annotated by Government Office Affairs Communication(GOAC) professional and it is labeled either positive and negative.
iii. Amharic Sentiment Lexicons: The Amharic sentiment lexicons includes manual (1000) (Gebremeskel, 2010), Amharic SentiWordNet(SWN)(13679) (Neshir Alemneh et al., 2019) and Amharic Semantic Orientation Calculator(SOCAL) (5683) (Neshir Alemneh et al., 2019). These lexicons are used as benchmark to compare the performance of PPMI based lexicons.

### 5.2 EXPERIMENTAL SETTINGS

Using the aforementioned corpus, Amharic News Corpus with 11433 documents and 2800 Facebook News post and Users Comments are used to build word-context PPMI. First, we tried to clean the data set. After cleansing, we tokenized and word-Context count dictionary is built. Relying on this dictionary and in turn, it is transformed into PPMI Sparse matrices. This matrix is saved in file for further tasks of Amharic Sentiment lexicon generation. The total vocabulary size is 953,406. After stop words are removed and stemming is applied, the vocabulary size is reduced to 231,270. Thus, the size of this matrix is (231270, 231270). Based on this word-word information in this matrix, the point-wise mutual information (PMI) of word-word is computed as in equation 2. The PMI is input to further computation of our corpus based lexicon expansion algorithm 1. Finally, we generated Amharic sentiment lexicons by expanding the seeds relying on the PPMI matrix of the corpus by implementing this algorithm 1 at two threshold iteration values of 100 and 200. With these iterations, we got corpus based Amharic Sentiment lexicons of size 1811 and 3794 respectively. We think, iterations >200 is better. Thus, our further discussion on evaluation of the approach is based on the lexicon generated at 200 iteration(i.e.Lexicon size of 3794). This lexicon saved with entries containing stemmed word, part of speech, polarity strength and polarity sign. Sample of this lexicon is presented in Table 2.

Table 2: Sample list of terms in the generated Amharic sentiment lexicon using PPMI with their corresponding surface words

| Stem | POS | Sentiment | Sample Surface Words |
|------|-----|-----------|----------------------|
| ሀስት /'cock and bull story'/ | noun | -0.82 | ['ሀስተኛ/lair/, ሀስት/fake/, የሀስት/fake/, ከሀስተኛ/from fake maker/, ሀስተኛና/lair and/, የሀስተኛ/for fake maker/, ለሀስተኛ/to lair/, ሀሴት/pleasure/, ታሀሳት//, ሀስነት/fakeness/, ሀስትን/fake/ , ከሀስተኛው/from the lair/, ሀስተኛው/lair/, ሀስቱን/lair/, ከሀስት/from lair/, ሀስትና/fake and/, ከሀሴቱ/from the fake/, ሀሴትና/pleasure and/, ሀስትን/the pleasure/, ሀሳት/fake/, ለሀስትና/to fake and/, ሁስት//, የሀስትና/for fake and/, ለሀስት/to fake/, ሀስቶች/alot of fake/,etc...] |
| ሀቅ / 'fact veracity'/ | Noun | +0.81 | [ሀቆች/facks/ ሀቅ/fact/, ሀቀኛ/honest/, ሀቅን/the truth/, ከሀቅ/from fact/, ሀቁ/the truth/, ሀቁን/the truth/, የሀቀኛ/the one who is honest/, ሀቁን/the truth/, ሀቀኛና/the truth and/, ሀቅና/truth and/, ከሀቁ/from the truth/, ሀቀኛውን/that who is honest/, ሀቀሚ//, ሀቆችን/facts/, ከሀቀኛ/from facts/, ሀቅነት/truthness/, የሀቅ/truth/, የሀቅ/for truth/, ሀቅና/fact and/, ለሀቅና/for truth and/, ለሀቅ/for truth/, የሀቅነት/for truthiness/, ሀቅ/fact/, ሀታዊ/honesty/, ሀቲ//, ስለሀቅ/for truth/, ሀቀኛው/the one who is honest/,etc...] |

We will evaluate in three ways: external to lexicon and internal to lexicon. External to lexicon is to test the usefulness and the correctness of each of the lexicon to find

sentiment score of sentiment labeled Amharic comments corpus. Internal evaluation is compute the degree to which each of the generated lexicons are overlapped( or agreed ) with manual, SOCAL and SWN(Amharic) Sentiment lexicons.

## 5.3 SUBJECTIVITY DETECTION

In this part , we evaluate the accuracy of the subjectivity detection rate of generated PPMI based Amharic lexicon on 2800 facebook annotated comments. The procedures for aggregating sentiment score is done by summing up the positive and negative sentiment values of opinion words found in the comment if those opinion words are also found in the sentiment lexicon. If the sum of positive score greater or less than sum of negative score, then the comment is said to be a subjective comment, otherwise the comment is either neutral or mixed. Based on this technique, the subjectivity detection rate is presented in Table 3.

Table 3: Comparison of the PPMI Lexicon' subjectivity detection rate (in%) on facebook comments

| Sentiment Lexicons | Detection withNoStem | Detection withStem |
|---|---|---|
| Amharic Manual(Baseline)[1] | 43.23 | 93.56 |
| Amharic PPMI(Ours) | - | 97.29 |
| Amharic SOCAL[2] | 31.28 | 96.65 |
| Amharic SWN [3] | 75.83 | 99.33 |

Discussion: As subjectivity detection rate of the PPMI lexicon and others are depicted in Table 3, the detection rate of PPMI lexicon performs better than the baseline(manual lexicon). Where as Lexicon from SWN outperforms the PPMI based Lexicon with 2% accuracy.

## 5.4 GENERATED LEXICON'S COVERAGE

This is to evaluate the coverage of the generated lexicon externally by using the aforementioned Amharic corpus(both facebook comments and general corpus). That is, the coverage of PPMI based Amharic Sentiment Lexicon on facebook comments and a general Amharic corpus is computed by counting the occurrence tokens of the corpus in the generated sentiment lexicons and both positive and negative counts are computed in percent and it presented in Table 4.

Table 4: Comparison of PPMI Lexicons' coverage (positive/negative count and in percent)

| Lexicons | 2500 Amharic Comments | | 20 millions Amharic tokens | |
|---|---|---|---|---|
| | coverage(+,-) count | in% | coverage(+,-) count | in % |
| Manual | [4399, 2995] | 31.95 | [2713167, 2161501] | 25.01 |
| SOCAL | [5738, 3953] | 41.87 | [4169817, 3391213] | 38.88 |
| SWN | [9447, 4803] | 61.57 | [6645592, 4006072] | 54.77 |
| PPMI | [5130, 4928] | 45.07 | - | - |

Discussion: Table 4 depicted that the coverage of PPMI based Amharic sentiment lexicon is better than the manual lexicon and SOCAL. However, it has less coverage than SWN. Unlike SWN, PPMI based lexicon is generated from corpus. Due to this reason its coverage to work on a general domain is limited. It also demonstrated that the positive and negative count in almost all lexicons seems to have balanced and uniform distribution of sentiment polarity terms in the corpus.

## 5.5 LEXICONS' AGREEMENT

In this part, we will evaluate to what extent the generated PPMI based Lexicon agreed or overlapped with the other lexicons. This type of evaluation (internal) which validates by stating the percentage of entries in the lexicons are available in common. The more percentage means the better agreement or overlap of lexicons. The agreement of PPMI based in percentage is presented in Table 5.

Table 5: The Agreement (in percent) between PPMI based lexicon and other Lexicons

| - | Lexicons Combinations | Agreement(in %) |
|---|---|---|
| 1 | PMI and SOCAL | 67.34 |
| 2 | PMI and SWN | 97.93 |
| 3 | PMI and Manual | 74.96 |

Discussion: Table 5 presents the extent to which the PPMI based lexicon is agreed with other lexicons.PPMI based lexicon has got the highest agreement rate (overlap) with SWN lexicon than the other lexicons.

### 5.6 Lexicon based Amharic Sentiment Classification

In this part, Table 6 presents the lexicon based sentiment classification performance of generated PPMI based Amharic lexicon on 2821 annotated Amharic Facebook Comments. The classification accuracy of generated PPMI based Lexicon and other lexicons are compared.

Table 6: The Accuracy (in percent) of PPMI based Lexicons for Sentiment Classification

| Amharic Lexicons | Accuracy(%) | | |
|---|---|---|---|
| | NoStem+NoNeg. | Stem+NoNeg. | Stem+Neg. |
| Manual(baseline) | *16.7* | *42.9* | 42.16 |
| PPMI | *-* | *-* | 48.87 |
| SOCAL | *14.6* | *46.3* | 47.2 |
| SWN | *30.9* | *50.1* | 48.87 |
| SOCAL +SWN | *443.7* | *66.6* | 70.26 |
| Manual+SOCAL +SWN | *53.7* | *75.8* | 78.19 |
| PPMI+SOCAL+SWN+Manual | ** | - | **83.51** |

Discussion: Besides the other evaluations of the generated PPMI based lexicon, the usefulness of this lexicon is tested on actual lexicon based Amharic sentiment classification. As depicted in Table 6 The accuracy of PPMI based lexicon for lexicon based sentiment classification is better than the manual benchmark lexicon. As discussed on dictionary based lexicons (Neshir Alemneh et al., 2019) for lexicon based sentiment classification in earlier section, using stemming and negation handling are far improving the performance lexicon based classification. Besides combination of lexicons outperforms better than the individual lexicon.

## 6 Conclusions

Creating a sentiment lexicon generation is not an objective process. The generated lexicon is dependent on the task it is applied. Thus, in this work we have seen that it is possible to create Sentiment lexicon for low resourced languages from corpus. This captures the language specific features and connotations related to the culture where the language is spoken. This can not be handled using dictionary based approach that propagates labels from resource rich languages. To the best of our knowledge, the the PPMI based approach to generate Amharic Sentiment lexicon form corpus is performed for first time for Amharic language with almost minimal costs and time. Thus, the generated lexicons can be used in combination with other sentiment lexicons to enhance the performance of sentiment classifications in Amharic language. The approach is a generic approach which can be adapted to other resource limited languages to reduce cost of human annotation and the time it takes to annotated sentiment lexicons. Though the PPMI based Amharic Sentiment lexicon outperforms the manual lexicon, prediction (word embedding) based approach is recommended to generate sentiment lexicon for Amharic language to handle context sensitive terms.

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
