# OpenReview forum: "Corpus Based Amharic Sentiment Lexicon Generation"
_ICLR.cc/2020/Conference — Reject_

### Official Review · AnonReviewer2 · 2019-10-23
**Official Blind Review #2**

**Rating:** 1

**Review:**

This work proposes a corpus based approach of mining lexicon for a low resource language focusing on Amharic sentiment. The proposed approach is a classic one already presented elsewhere, i.e., PMI to compute the similarity of terms using the co-occurrence statistics and thresholding to control the precision of lexicon.

I have strong concerns to this work.

- The proposed method already appeared in many literatures under different tasks, e.g., translation lexicon generation, or languages, e.g., Chinese, and the proposal is merely a yet another application of PMI.

- The condition of the proposed method is not clear. I initially assumed an existence of a sentiment dictionary for Amharic, but section 3 introduced other resources, e.g., English/Amharic dictionary, which are not clearly stated in the introductory nor in the beginning of section 3. I'd strongly suggest the author to make it clear what kind of assumptions are made, e.g., existence of bilingual dictionary, or not. Otherwise, it is very hard to replicate the experiments.

- Baselines sound not appropriate. Authors should conduct additional studies, e.g., graph propagation algorithm, to make sure that the proposed heuristic based approach is superior or not.

I think this work should have been submitted to other conferences, workshops or journals, focusing more on low resource languages.

**Experience Assessment:**

I have published one or two papers in this area.

**Review Assessment: Checking Correctness Of Derivations And Theory:**

I assessed the sensibility of the derivations and theory.

**Review Assessment: Checking Correctness Of Experiments:**

I assessed the sensibility of the experiments.

**Review Assessment: Thoroughness In Paper Reading:**

I read the paper at least twice and used my best judgement in assessing the paper.

---

### Official Review · AnonReviewer1 · 2019-10-23
**Official Blind Review #1**

**Rating:** 1

**Review:**

This paper introduces a corpus-based approach to build sentiment lexicon for Amharic. In order to save time and costs for the resource-limited language, the lexicon is generated from an Amharic news corpus by the following steps: manually preparing polarized seed words lists (strongly positive and strongly negative), calculating the co-occurrence of target word in its context via Positive Point-wise Mutual Information (PPMI) method, measuring the similarity between target words and seed words by cosine distance, iterating with the threshold 100 and 200. The PPMI lexicon is stemmed and evaluated from aspects of subjectivity detection, coverage, agreement and sentiment classification. Three other lexicons: Manual developed by manual, SOCAL and SWN developed by bilingual dictionary, are used as benchmark to compare with the PPMI lexicon. In sentiment classification experiment the PPMI lexicon did not show a superior performance. All the four lexicons have similar accuracy, between 42.16% ~ 48.87%.  Only when the four are combined together the result is improved to 83.51%.

While this paper presents an economical and practical method to generate a sentiment lexicon for resource-limited language it is not acceptable in it's current state to ICRL. The following points should be improved or clarified.
(1) The generalizability to any language needs to be shown. What are the lessons learned for any resource-limited language?
(2) The fit to ICLR is not perfect. I would expect a stronger focus on representation learning.
(3)	The conclusion is not well proved. Especially about the claim that their method is “with almost minimal costs and time”. PPMI lexicon might cost less than Manual lexicon, but SWN and SOCAL (Neshir Alemneh et al., 2019) are automatically generated by English-Amharic dictionary, not manually. Their generation costs and time are not told and can not be compared here.
(4)	The paper is incorrectly structured. The part “4 RESULTS AND DISCUSSION” presents neither results nor discussion. Part 4.1 shows seed words which should belongs to part 3. Part 4.2 shows stemming which should belong either to part 3 or to part 5. Therefore, it is better to make a restructure.
(5)	The contents following “Discussion:” in part 5 are not discussion.  Moreover, there is almost no discussion in this paper.
(6)	In part 5.4 Table 4, why there is result of PPMI in column “2500 Amharic Comments” but no result in column “20 million Amharic tokens”? Are the 2500 comments stemmed?
(7)	In part 5.6, why negation handling is adopted in part 5.6?

Minor comments:
(1)	Grammatical problems such as: Page 3 last paragraph: “Amharic is highly morphological language both inflectional and derivational morphology is complex.”
(2)	Page 6 last paragraph: “We will evaluate in three ways: external to lexicon and internal to lexicon.” It should be two ways.
(3)	The number of facebook comments varies: it is 2800 in part 5.3, 2500 in 5.4 including Table 4, and 2821 in 5.6.


**Experience Assessment:**

I have published one or two papers in this area.

**Review Assessment: Checking Correctness Of Derivations And Theory:**

I assessed the sensibility of the derivations and theory.

**Review Assessment: Checking Correctness Of Experiments:**

I assessed the sensibility of the experiments.

**Review Assessment: Thoroughness In Paper Reading:**

I read the paper at least twice and used my best judgement in assessing the paper.

---

### Official Review · AnonReviewer3 · 2019-10-27
**Official Blind Review #3**

**Rating:** 1

**Review:**

This paper proposes a domain-specific corpus-based approach for generating semantic lexicons for the low-resource Amharic language. Manual construction of lexicons is especially hard and expensive for low-resource languages. More importantly, the paper points out that existing dictionaries and lexicons do not capture cultural connotations and language specific features, which is rather important for tasks like sentiment classification. Instead, this work proposes to automatically generate a semantic lexicon using distributional semantics from a corpus.

The proposed approach starts from a seed list of sentiment words in 3 pre-determined POS classes. This is followed by deriving a PPMI matrix from the word-context co-occurence matrix (context size= 2). Now given a word, the cosine distance is computed from centroid of each seed classes and words which are similar than a given threshold, are added to the original list. This process is repeated for a pre-specified number of iterations. They apply the generated lexicons to subjectivity detection and sentiment classification task in a small annotated Amharic corpus of facebook posts

Strengths:

1. NLP tasks on low-resource language is very challenging. This paper proposes a efficient, unsupervised way of gathering semantic lexicons that perform reasonably well on a downstream task.

Weaknesses:

1. Related work: The paper needs to cite and mention a much more broader literature. Currently, it mentions just 3 related work, and contrasts itself with one. But given there is so much work on distributional semantics (in English and other languages), they deserve mentioning. Few example of ones are 1. The distributional inclusion hypotheses and lexical entailment (Geffet and Dagan 2005), Distributed representations of words and phrases and their compositionality (Mikolov et al., 2013), Improving hypernymy detection with an integrated path-based and distributional method (Shwartz et al., 2017), Distributional Inclusion Vector Embedding for Unsupervised Hypernymy Detection (Chang et al., 2018). However, this is by no means a complete list and you should refer to these papers to get a list of other relevant work.
2. Novelty of work: The novelty of the work is rather limited and the paper should try more low-dimensional embedding based approaches which has been proven to be very effective for a wide variety of tasks. In section 3, the paper mentions there are primarily two kinds of approaches (count based vs embedding based) — The paper should motivate why it chose one over another.
3. Baselines: The paper would benefit from having some comparison with learned baselines trained from some distantly supervised data from, for example the SWN and SOCAL lexicons.
4. Organization of the paper: The writing and the organization of the paper needs to be better. For example, equation 3, 4, 5 are rather simple and can be condensed to one equation (no need to show averaging of the seed embeddings). Also in equation 3, would \vec{w} would be \vec{x}?. The results and discussions did not have either and over-all there are many grammatical errors.
5. I am not sure what the subjectivity detection task is in Table 3. Is it a standard task -- if not, the paper should define it first.

Overall, the paper is a nice effort but in its current form it is not ready for ICLR and I hope the comments will help make the paper better for upcoming NLP workshops and conferences.

**Experience Assessment:**

I have read many papers in this area.

**Review Assessment: Checking Correctness Of Derivations And Theory:**

N/A

**Review Assessment: Checking Correctness Of Experiments:**

I assessed the sensibility of the experiments.

**Review Assessment: Thoroughness In Paper Reading:**

I read the paper at least twice and used my best judgement in assessing the paper.

---

### Decision · Program_Chairs · 2019-12-19

**Decision:**

Reject

**Comment:**

This papers addresses the problem of creating sentiment lexicon for a resource limited language (Amharic). This task is time consuming and requires skilled annotators. Hence the authors propose a method for constructing this automatically from News corpora. They start with a seed list of sentiment bearing words and then add new words to this list based on their PPMO scores with existing words.

While the reviewers agreed that this work is of practical importance, they had a few objections which I have summarised below:
1) Lack of novelty: The work has very few new ideas
2) Lack of comparison with existing work: Several missing citations have been pointed out by the reviewers
3) Weak experiments: The experimental section needs to be strengthened with more comparisons to existing work as well as proving the results for at least one more language.
4) Organisation of the paper: The paper needs to be restructured for better presentation. In particular,  the Results and Discussions section does not really contain any discussions.
5) Grammatical errors: Please proofread the paper thoroughly and fix all grammatical and typo errors.

Based on the reviewer comments and lack of any response from the authors, I recommend that the paper in it current form cannot be accepted.